

# Maintenance of dominant populations in heavily grazed grassland: Inference from a *Stipa breviflora* seed germination experiment

Wenting Liu[1], Zhijun Wei[2] and Xiaoxia Yang[1]

[1] Qinghai University, Qinghai Academy of Animal and Veterinary Sciences, Xining, China
[2] Key Laboratory of Grassland Resources, Ministry of Education P.R. of China, Huhhot, P.R. China, College of Grassland, Resources and Environment, Inner Mongolia Agricultural University, Hohhot, China

## ABSTRACT

An understanding of population adaptation and maintenance mechanisms under interference from large herbivores is lacking and is a major focus of ecological research. In the Eurasian steppe, which has been subjected to continuous interference from domesticated ungulates throughout history and shows increased grazing, it is particularly urgent to analyze the ecological adaptation strategies of widely distributed *Stipa* plants. In this study, *Stipa breviflora* in a group of desert steppes in the Mongolian Plateau was selected to study the potential mechanism underlying the maintenance of dominant populations under the continuous interference of heavy grazing from the new perspective of seed germination rate. Laboratory experimental results showed that the values of the phenotypic traits of *S. breviflora* seeds were lower under a heavy grazing treatment than under a non-grazing treatment, but the seed germination rate did not decrease. The awns of non-grazed seeds significantly affected the seed germination rate, while those of heavily grazed seeds did not. Field observations showed that grazing does not significantly affect the population density of *S. breviflora* at different growth stages except in extremely wet and dry years. Our study suggests that under heavy grazing, *S. breviflora* uses an "opportunistic" ecological strategy to ensure population maintenance by increasing the seed germination rate and reducing dispersal via changes in associated seed phenotypic traits.

## INTRODUCTION

The Eurasian steppe, which includes a wide distribution of *Stipa* plants (*Gonzalo, Aedo & García, 2013*; *Ghiloufi, Büdel & Chaieb, 2016*; *Nobis et al., 2016*; *Wieczorek et al., 2017*), stretches across Northeast China, Mongolia, Russia, Ukraine and Hungary. Historically, this region has been subjected to continuous interference from domesticated ungulates, and grazing has been increasing, leading to widespread declines in species

Corresponding author
Zhijun Wei, nmndwzj@imau.edu.cn

diversity and above-ground productivity (*Bai et al., 2007*). However, even in desert steppes with average annual precipitation of less than 200 mm and long-term overgrazing, it is still possible to find *Stipa* plants living stably and with relatively high biomass (*Wang et al., 2014*). Therefore, explaining the ecological strategies adopted by the *Stipa* genus is essential for the conservation management of plant species and our understanding of species dynamics and maintenance mechanisms.

The selective feeding theory for livestock is obviously unable to explain the survival strategies of plants under severe interference from herbivores. The passive, indiscriminate and repeated feeding of herbivores destroys the vegetative and reproductive organs of plants and causes negative effects such as leaf shortening and narrowing and cluster shrinking (*Díaz et al., 2007*). Life history theory posits that plants have substantial flexibility in resource allocation between vegetative and reproductive growth (*Miller, Tyre & Louda, 2006*). To ensure the survival of the plant, the vegetative organs are often favored at the expense of sexual propagation. Assuming that this compensation continues, the survival of plant species will be difficult. Therefore, we expect that the species that can survive stably will have a set of effective ecological strategies allowing them to overcome this dilemma, and sexual propagation is the key link.

*Louault et al. (2005)* compared traits in long-term grazed grassland with those in non-grazed grassland and found that grazing caused a series of effects, such as a reduction in seed size and a decrease in seed weight. The size of the seed directly affects its scattering, spread and survival (*Chambers, 1995*). Compared with large-sized seeds, small-sized seeds are able to disperse farther, thereby improving their fitness (*Coomes & Grubb, 2003*). Studies have found that the awn of *Stipa* seeds can help them spread more efficiently. For example, the pubescence on the awns erects when the air is dry to increase the aerostatic buoyancy (*Peart, 1981*), which reduces the speed at which the seed falls (*Greene & Johnson, 1993*); in addition, the pubescence on the awns can attach to the fur of animals to spread the seed. When the propagule falls to the ground, the moisture absorption by the awn allows the seed to move a short distance on the soil surface and complete the self-burial process, allowing the propagule to enter the soil (*Elbaum et al., 2007*; *Jung, Kim & Kim, 2014*; *Liu et al., 2018b*). All of the abovementioned studies suggest the importance of awns for the spread of seeds. There are two interesting conjectures. (Conjecture 1) Awns can inhibit the germination of seeds. Assume that the dispersal process of seeds has not been completed, and seed germination occurs because of favorable temperature, humidity, etc. This seed germination strategy is obviously not ideal. (Conjecture 2) Under heavy grazing, awns cannot effectively inhibit seed germination to prevent unfavorable seed germination and increase the chance of population regeneration.

Desert grassland, accounting for 39% of the total grassland area in northern China, is the most arid grassland ecosystem on the Eurasian steppe (*Li et al., 2000*). This grassland is an important part of the semi-arid grassland area of northern China and has a long history of animal husbandry (*Wang et al., 2017b*). Compared with other grassland types, desert grassland has a low plant species richness and primary productivity due to drought caused by limited rainfall (*Li et al., 2008*). Therefore, it is valuable to analyze the

survival strategies of the dominant species of the desert steppe that have been overgrazed. Therefore, this study aims to examine the following by focusing on *Stipa breviflora* in the desert grassland in northern China: (1) the effect of grazing on the morphological characteristics of seeds; (2) the effect of awns on seed germination under grazing and non-grazing conditions; and (3) the interannual variability in plant population density in grazing and non-grazing treatments.

## MATERIALS AND METHODS

### Study area

The experimental area is located in the Inner Mongolia Plateau *S. breviflora* positioning station of Inner Mongolia Agricultural University, Zhurihe Town (E112°47′16.9″, N42°16′26.2″), Xilin (Fig. 1), with an elevation of 1,100–1,150 m. Here, rain and hot temperatures occur in the same period, with average annual precipitation of 183.0 mm, annual average temperature of 5.8 °C, average annual sunshine hours of 3,137.3 h, average annual evaporation of 2,793.4 mm, and average annual frost-free period of 177 days. The area is characterized by a medium-temperate climate. The average wind speed is 5.1 m/s, and the wind is mostly concentrated in winter and spring and dominated by wind from the northwest. The average number of windy days is 67, and dust storms frequently occur. The soil is light chestnut soil with a deserted surface, and the humus layer is 20–30 cm thick. The grass layer is low, generally 10–25 cm in height, and has a low coverage of 15–25%. The vegetation is composed of *S. breviflora* (cluster of a perennial grass); *Cleistogenes songorica* and *Allium polyrhizum* are the dominant plants.

### Experimental design

The grazing trial began in 2010, starting in May and ending in late October each year. During the experimental period, continuous grazing was adopted. In the evening, the flock was allowed in the plot, with no supplementary feeding. The test plots selected in this study are located in the same continuous section with flat terrain and a relatively uniform environment, effectively controlling for differences in background, terrain, and spatial heterogeneity. There were two treatments in this study, namely, the heavy grazing and non-grazing treatments, each with three replicates, and each test plot had an area of 2.60 ha. The heavy grazing treatment had a stocking capacity of 3.08 sheep·ha$^{-1}$·a$^{-1}$, and the non-grazing treatment excluded interference from humans, livestock, and large wild animals (*Liu et al., 2018a*). The herbivores in the grazing area were Sunite sheep that had generally the same health status, individual size, weight, and sex.

### Seed germination experiment

Since the seeds of *Stipa* undergo a period of dormancy after maturation (*Zhang et al., 2017*), this study attempts to explore the ecological strategies of *S. breviflora* populations under grazing from the perspective of seed germination. To minimize the effect of seed dormancy, we chose the period when the germination rate of *S. breviflora* seeds was the highest. Then, on June 3, 2015, in each experimental plot, mature reproductive

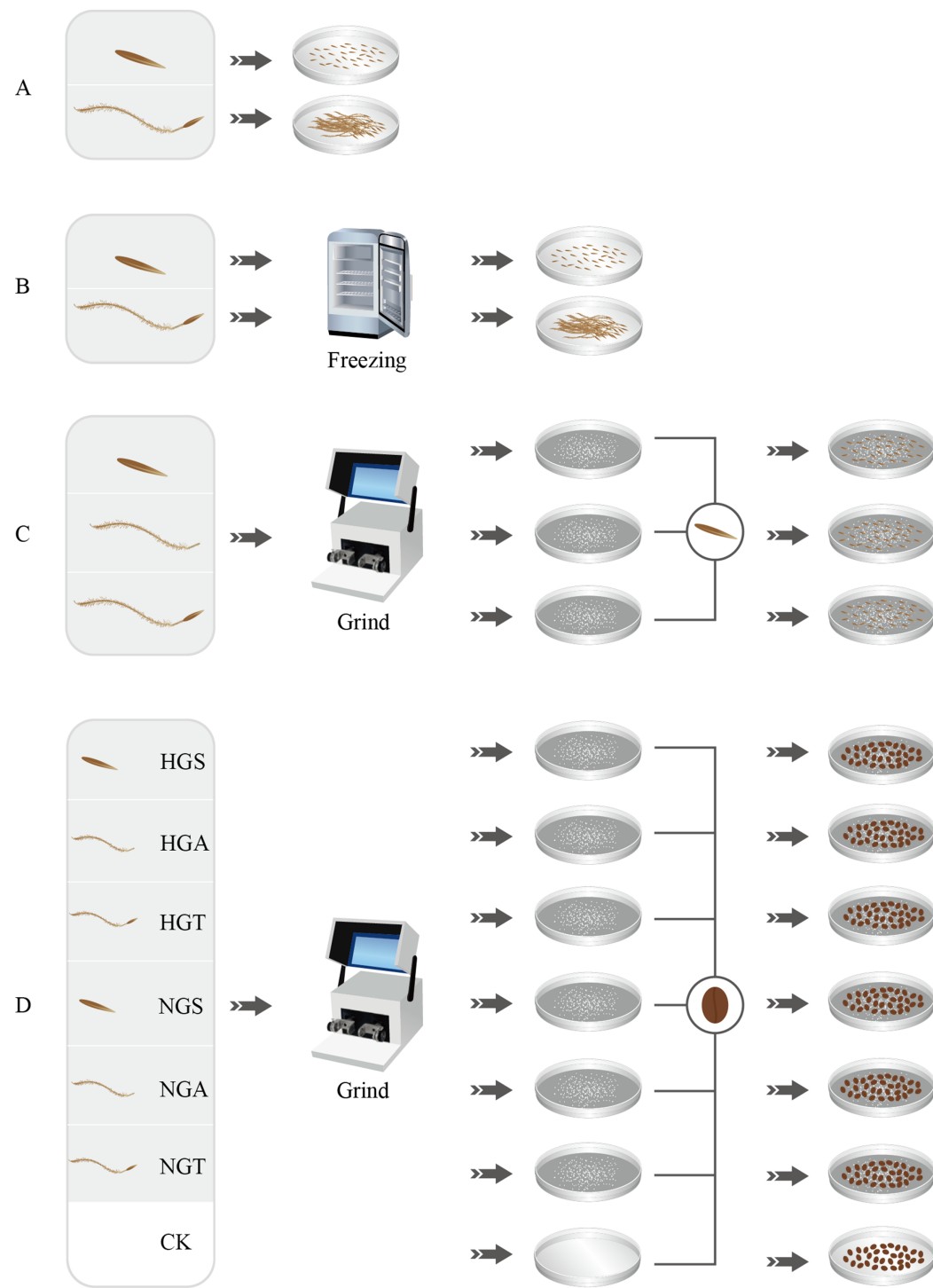

**Figure 1 Seed germination test process.** A, B, C and D represent four germination experiments, respectively. HGS, HGA, HGT, NGS, NGA, NGT and CK represent the heavily grazed seed, heavily grazed awn, heavily grazed seed + awn, non-grazed seed, non-grazed awn, and non-grazed seed + awn and control treatment, respectively.

shoots of strong *S. breviflora* plants with a high seed-setting rate were selected. After being air dried, the seeds were manually threshed and stored in a bag for winter use. On June 15, 2016, the seeds were brought back to the laboratory, and the seeds from the same treatment were uniformly mixed. A total of 50 seeds were randomly selected for each grazing treatment, and the seed width, seed length, seed weight, and awn weight were measured with Vernier callipers.

(1) Seeds with superior morphological characteristics were selected, rinsed with tap water, sterilized with 75% ethanol, rinsed with deionized water, and treated with awning and non-awning treatments. The seeds were then placed in 20 nine-cm-diameter Petri dishes, with 50 seeds per dish. Each treatment was repeated five times. The light and dark periods were each 12 h, and the dishes were kept at 25 °C in a constant temperature incubator. The germination status of the *S. breviflora* seeds in each dish was recorded every 24 h. An appropriate amount of water was added after recording the germination status to ensure consistent processing conditions. The assay was terminated after 14 days (*Liu et al., 2018b*) (Fig. 1A).

(2) To assess whether the seeds of *S. breviflora* were dormant, a low-temperature treatment method was used in this study as well as in preliminary and related studies (*Wang et al., 2017a*). The seeds from the heavy grazing treatment and non-grazing treatment were placed in a refrigerator at 4 °C for the low-temperature treatment. After 4 weeks of the low-temperature treatment, all seeds were removed, rinsed with distilled water and then germinated. Step 1 was repeated for the germination test (Fig. 1B).

(3) To analyze the potential factors affecting the germination rate (germination percentage) of *S. breviflora* seeds, we added the awns, seeds, and seeds + awns, which were heavily grazed, to the culture medium for the germination test of non-grazed seeds (NGSs), whose awns, seeds, and seeds + awns were separately added to the culture medium for the germination test of heavily grazed seeds. Specifically, 50 heavily grazed *S. breviflora* seeds (HGSs) were selected and pulverized with a ball mill, and the powder was then placed in culture medium and covered with damp blotting paper. The production of heavily grazed awn (HGA), heavily grazed seed + awn (HGT), NGS, non-grazed awn (NGA), and non-grazed seed + awn (NGT) additives was completed as described above. The seed germination test without additives was set as a control treatment (CK), and each treatment was repeated five times. A total of 50 heavily grazed seeds were placed in the medium where NGS additives had been added, and 50 NGSs were added to the medium where heavily grazed seed additives had been added; an appropriate amount of distilled water was then supplied. Step 1 was repeated for the germination test (Fig. 1C).

(4) To further verify the factors affecting the germination rate of *S. breviflora* seeds, this study selected cabbage seeds as the test object and used the six types of additives described above to perform germination tests (*Bao et al., 2015*). Germination experiments without any additives were set as controls, and each treatment was repeated five times. A total of 50 grazed cabbage seeds were placed in each medium, and the seed germination process was carried out with the abovementioned germination test procedure (Step 1) (Fig. 1D).

## Survey of the *S. breviflora* population

During the period of peak plant growth from 2010 to August 2017, five sample plots, each with an area of $1 \times 1$ m$^2$, were randomly selected in each test plot to determine the number of *S. breviflora* individuals. According to the measured basal diameter (Bd) of *S. breviflora* and a previous study (*Liu et al., 2017*), *S. breviflora* was divided into five stages based on age: seedling stage (Bd $\leq$ 4 mm), juvenile stage (4 mm < Bd $\leq$ 20 mm), adult stage (20 mm < Bd $\leq$ 40 mm), pre-ageing stage (40 mm < Bd $\leq$ 70 mm), and ageing stage (Bd > 70 mm). From 2015 to the first 10-day period of August 2017, five sample plots, each with an area of $1 \times 1$ m$^2$, were randomly selected in each test plot to investigate the number of *S. breviflora* individuals at each growth stage.

## Meteorological data

Meteorological data were obtained from a micro weather station (GroWeather® software version 1.2; Davis Instruments Corporation, Hayward, CA, USA) located at the experimental site (*Wang et al., 2014*).

## Statistical analysis

An independent *t*-test was conducted to study the seed width, length and weight; awn weight; germination rate of seeds with awns and those without awns as well as the germination rate of seeds with awns and those without awns after the non-grazing treatment under the heavy grazing and non-grazing treatments. One-way analysis of variance was performed to compare the NGSs, NGAs, NGTs and germination rates of heavily grazed *S. breviflora* seeds with the CK, and Duncan's test was used for multiple comparisons. Furthermore, one-way analysis of variance was performed to compare the HGSs, HGAs, HGTs and germination rates of non-grazed *S. breviflora* seeds with the CK, and Duncan's test was used for multiple comparisons. Additionally, the NGSs, NGAs, NGTs, HGSs, HGAs, HGTs and seed germination rates of Chinese cabbage (CK) were analyzed, and Duncan's test was used for multiple comparisons. An independent *t*-test was adopted to compare the density ratio of *S. breviflora* and the density of *S. breviflora* between the heavy grazing treatment and non-grazing treatment in the same year. The statistical analyses described above were completed in SAS 9.0 (SAS Institute Inc., Cary, NC, USA).

# RESULTS

## Seed phenotypic traits and germination rate

The results of the independent *t*-test (Fig. 2) showed that the values of the phenotypic traits of non-grazed *S. breviflora* seeds were significantly higher than those of heavily grazed *S. breviflora* seeds ($P < 0.001$); specifically, the seed length, width, and weight and awn weight were 10.47%, 13.60%, 36.02%, and 41.85% higher, respectively. The germination rate of non-grazed *S. breviflora* seeds with no awns was 386.96% higher than that of such seeds with awns ($P < 0.001$), while there was no significant difference in the germination rate between the two types of seeds under the heavy grazing treatment ($P > 0.05$) (Fig. 3A). The germination rate of heavily grazed seeds with awns

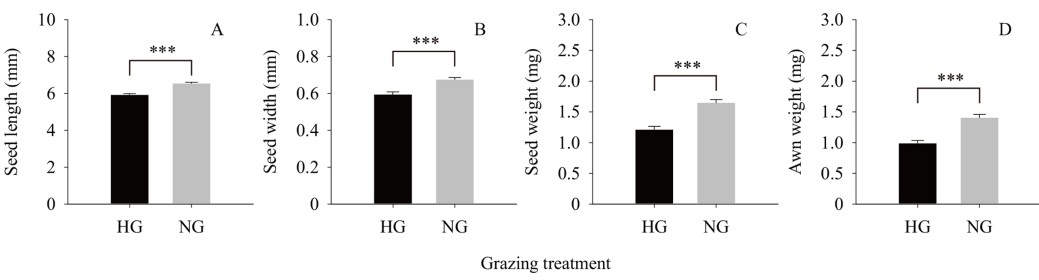

**Figure 2 Seed characteristics of *Stipa breviflora*.** A, B, C and D represent seed length, seed width, seed weight, awn weight , respectively. *** means *P* < 0.001.

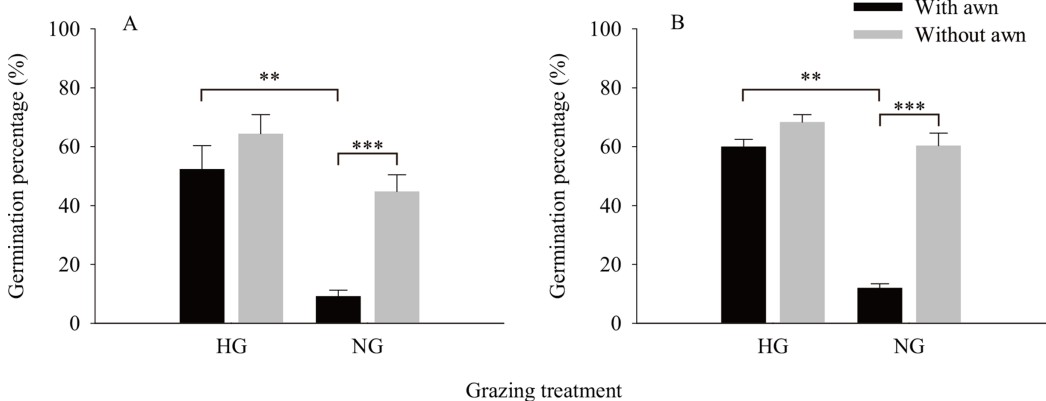

**Figure 3 Characteristics of seed germination of *Stipa breviflora*.** A and B represent germination experiment (1) and (2), respectively. ** means *P* < 0.01, *** means *P* < 0.001.

was significantly higher than that of NGSs with awns. However, there was no significant difference in the germination rate of seeds with no awns between the heavy grazing treatment and the non-grazing treatment.

## Impact of awns on seed germination rate

Regarding whether the germination rate of *S. breviflora* seeds is affected by seed dormancy, this study found that the germination rate was the same before and after dormancy release by physically breaking dormancy. Specifically, under the non-grazing treatment, seeds with awns had a significantly lower germination rate than those without awns, and the germination rate of seeds without awns was not obviously different between the heavy grazing treatment and the non-grazing treatment (Fig. 3B).

The results of the tests with additives showed (Fig. 4) that *S. breviflora* seeds under the heavy grazing treatment in the culture medium with the NGA and NGT additives had a significantly lower germination rate than seeds with no additive (CK) (*P* < 0.05), while the germination rate of those with NGS additives did not significantly differ from that of seeds with no additive (CK) (*P* < 0.05) (Fig. 4A). In the culture medium of non-grazed *S. breviflora* seeds, the germination rate with the NGS additive was significantly higher (*P* < 0.05) than that with the NGA additive under heavy grazing,

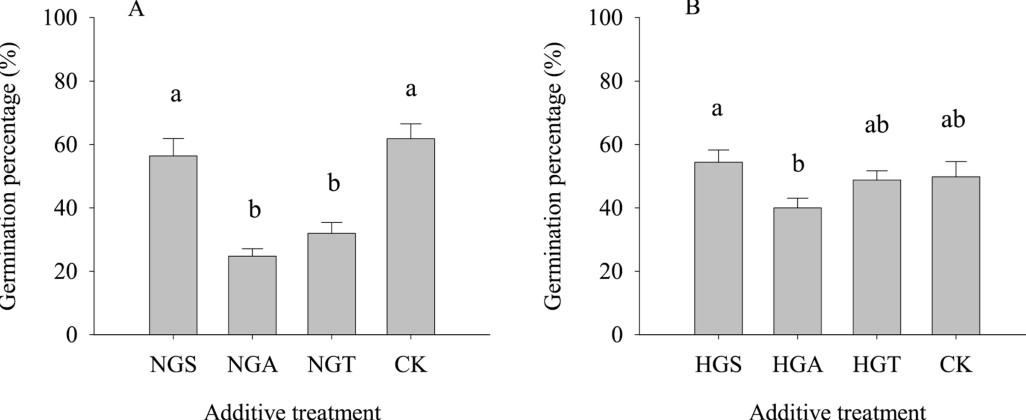

**Figure 4 Effect of *Stipa breviflora* additives on the seed germination of *Stipa breviflora*.** A, the seed germination rate of Stipa breviflora treated with heavy grazing after adding non-grazing seeds (NGSs), awns (NGAs), and seeds + awns (NGTs) was studied. B, the germination rate of Stipa breviflora seeds (HGSs), awns (HGAs), and seeds + awns (HGTs) without grazing was studied. CK denotes the control. Different lowercase letters indicate significant differences, $P < 0.05$.

but there was no significant difference in the germination rate between those with the NGT additive and those without any additive (Fig. 4B).

To avoid the regulation of *S. breviflora* seed germination by their own substances and further verify that awns affect the germination of *S. breviflora* seeds, this study used cabbage seeds as a test object. The germination rate of Chinese cabbage seeds with the awn additive was significantly lower than that of the CK, and that of cabbage seeds with the NGA additive was 49.10% lower than that under the heavy grazing treatment ($P < 0.05$) (Fig. 5).

## Dynamics of *S. breviflora* populations

The 2010–2017 field survey results (Fig. 6) showed that in all years except 2014 and 2016, there was no significant difference in the ratio of the density of heavily grazed *S. breviflora* plants to the number of plants in the community and that of non-grazed *S. breviflora* plants to the number of plants in the community ($P > 0.05$). Further analysis revealed that there was also no significant difference in the density and density ratios of *S. breviflora* at different growth stages between the heavy grazing and non-grazing treatments in 2015 and 2017, and in 2016, the *S. breviflora* seedlings and juveniles in the non-grazing treatment showed a significantly higher density than those in the heavy grazing treatment, with no difference displayed at any of the other growth stages (Fig. 7).

## DISCUSSION

*Chen et al. (2017)* found that heavy grazing affects the morphological characteristics of grassland plant seeds, and this finding is consistent with the findings of this study. Specifically, compared with the non-grazing treatment, heavy grazing significantly reduced the seed length, seed width, seed weight, and awn weight of *S. breviflora* on the desert steppe. This finding may be because grazing reduced the number of reproductive individuals in the *S. breviflora* population (*Liu et al., 2018b*). Moreover, although the plants

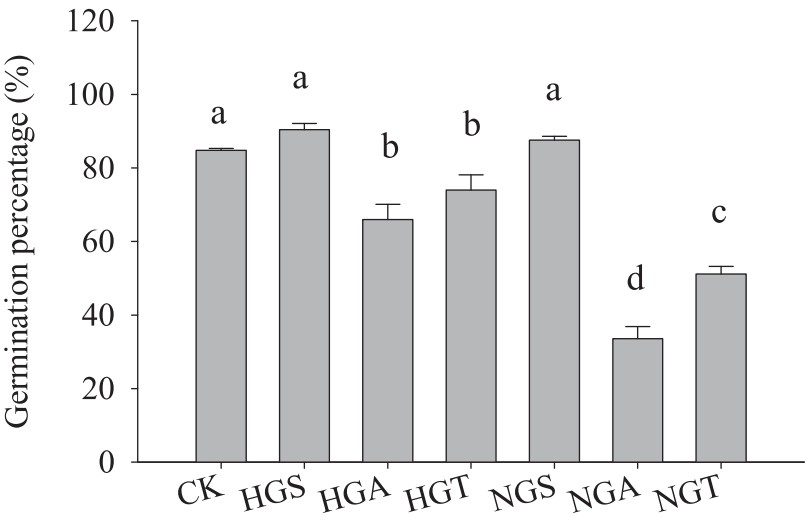

**Figure 5** **Effect of *Stipa breviflora* additives on the seed germination of Chinese cabbage.** HGS, HGA, and HST indicate the seed, awn and seed + awn of *Stipa breviflora* under heavy grazing. NGS, NGA, and NGT indicate the seed, awn, and seed + awn of *Stipa breviflora* under no grazing. CK was treated for contrast. Different lowercase letters indicate significant differences, $P < 0.05$.

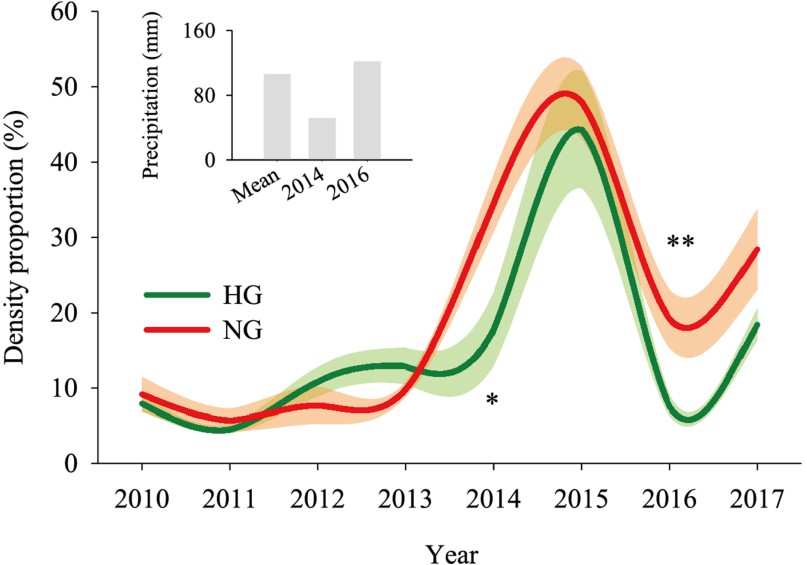

**Figure 6** **Population dynamics of the *Stipa breviflora* population.** The shadow depicts the standard error. *Denotes $P < 0.05$; **indicates $P < 0.01$. HG and NG represent heavy grazing and no grazing, respectively.

have substantial flexibility in resource allocation between growth and reproduction (*Miller, Tyre & Louda, 2006*), their vegetative and reproductive organs are destroyed when they suffer from severe interference from herbivores and repeated feeding by livestock. When plants have a constant total amount of resources and a limited ability to assimilate

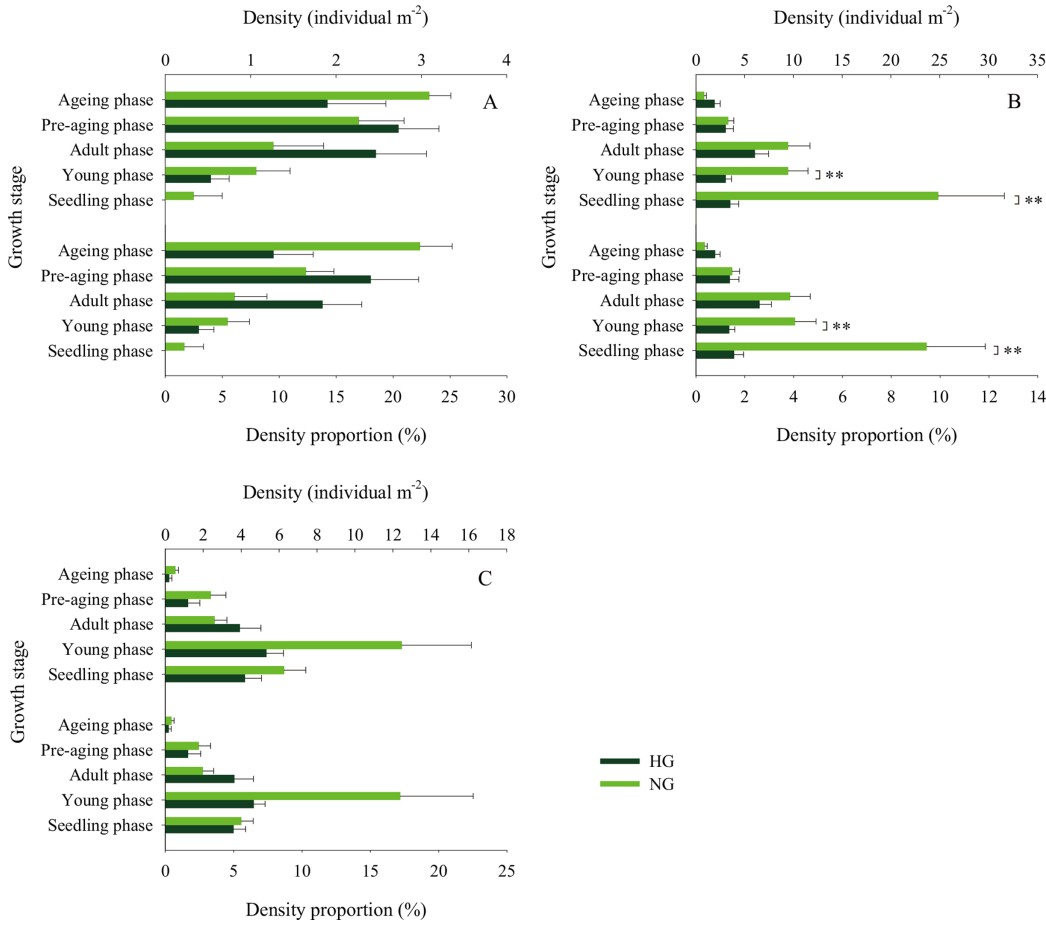

**Figure 7 Population density characteristics of *Stipa breviflora* at different growth stages.** (A) 2015, (B) 2016, (C) 2017. **indicates *P* < 0.01. HG and NG represent heavy grazing and no grazing, respectively.

resources, they have to invest more resources to produce vegetative organs, compensating for the grazing-induced damage at the expense of sacrificing sexual reproduction (*Fornara & Du Toit, 2007*) and causing a series of reactions such as the seed shortening, narrowing, and weight reduction observed in *S. breviflora*. A study in the Mediterranean steppe further validated our conjecture. When the stocking capacity is greater than three sheep·ha⁻¹, the native plant species show a significant increase in vegetative growth and a reduction in reproductive growth (*De Miguel et al., 2010*). This pattern indicates that to some extent, phenotypic trait changes are a comprehensive manifestation of plant populations adapting to changes in the external environment (*Louault et al., 2005*).

Although heavy grazing alters the morphological characteristics of seeds, it is interesting to note that heavily grazed *S. breviflora* seeds (without an awn) generally exhibit the same germination rate as NGSs (without an awn); that is, the germination rate of large seeds does not differ from that of small seeds, which is contrary to the results of *Zimmerman & Weis (1983)*, *Hendrix & Trapp (1992)*, and *Greipsson & Davy (1995)*. We assume that in response to heavy grazing, *S. breviflora* has formed its own unique survival strategy over its long-term interaction with livestock: *S. breviflora* can carry out

both sexual reproduction and asexual reproduction. However, it is obviously not favorable to increase the number of asexually reproducing individuals under severe interference from livestock because individuals who vegetatively propagate seedlings usually form small-scale aggregations with the mother plant, which will undoubtedly reduce the feeding cost of livestock and is not conducive to the survival of the population. Compared to large seeds (no grazing), at the same wind speed, awns can help smaller seeds (heavy grazing) disperse farther (*Matlack, 1987*), and this increase in dispersal is more propitious for the spread and colonization of the population. However, there is an obvious problem. If the seed can spread farther but has a lower germination rate, then further dispersal obviously has no biological significance. This condition means that the seeds of heavily grazed plants need a higher germination rate. Our previous findings provide the basis for this hypothesis. Grazing can disrupt the threshold for germination of *S. breviflora* seeds; that is, the weights of germinable seeds under a heavy grazing treatment were 25% lower than those under a non-grazing treatment (*Liu et al., 2018b*). The results of this study also showed that grazing does not affect the germination rate of *S. breviflora* seeds (Fig. 3), suggesting that the high germination rate of heavily grazed seeds may be an important mechanism for population maintenance.

Additionally, the germination rate of heavily grazed *S. breviflora* seeds (with awns) was significantly higher than that of NGSs (with awns), and the germination rate of awn-free seeds was significantly higher than that of seeds with awns. This result indicates that awns can dramatically affect the germination rate of *S. breviflora* seeds. Numerous studies have suggested that rapid seed germination is a common survival mechanism for plants in arid regions (*Wallace, Rhods & Frolich, 1968*), but our results do not fully support this view. This discrepancy probably occurs because the desert grassland is dry and rainless, and effective rainfall can promote not only large-scale seed germination but also the growth and development of individual plants. In the non-grazing treatment, with relatively good plant growth, this increase in performance due to rainfall will undoubtedly increase the competitiveness within and among plant species and cause the death of *S. breviflora* seedlings that are poor competitors, which is not conducive to the regeneration and survival of the population. Furthermore, awns can help seeds disperse over long distances. If seeds germinate before they are dispersed, then this trait is obviously not propitious for the colonization and reproduction of the population. *Garnier & Dajoz (2001)* also confirm our view, as the germination of *Hyparrhenia diplandra* seeds was not observed during the dispersal process in savannas. All of the abovementioned findings indicate that awns can reduce the germination rate of *S. breviflora* seeds, especially those not subjected to grazing (Fig. 5). Therefore, we believe that the awn of the propagule may play an extremely important role in the reproduction of plants. Once the seeds are mature, the propagule is detached from the spikelets by the twisting of the awns (*Raju & Ramaswamy, 1983*), and the population is able to spread with the help of wind or animals. In this process, the presence of the awn columns inhibits the germination of the seeds and prevents germination from occurring under unfavorable conditions.

However, in the heavy grazing treatment, although the germination rate of seeds without awns was 22.90% higher than that of seeds with awns, the awns did not

significantly affect the germination rate of *S. breviflora* seeds (Fig. 3). This result seems to contradict the aforementioned result, but we believe that this pattern reflects a series of behavioral mechanisms for plants to adapt to herbivores. The change in awns is highly heritable (*Garnier & Dajoz, 2001*). In the case of heavy grazing, due to long-term, intense feeding of livestock, individual plants become sparse, the competition within and among species is weakened, the exposed land area is increased, and the water holding capacity is reduced. Assuming awns effectively inhibited seed germination at this time, this would undoubtedly exacerbate the decline in the population. Therefore, under heavy grazing conditions, the presence of awns not only provides more favorable sites for the survival of new individuals but also does not force the seeds to germinate after awning, which greatly reduces the time required for seed germination and creates greater odds for the survival of the *S. breviflora* population.

Based on the population density of *S. breviflora* expressed as a proportion of the number of plants in the community in 2010–2017, our research verifies the abovementioned theory to a certain extent; that is, there was no significant difference between the heavy grazing treatment and non-grazing treatment (except in 2014 and 2016). *Oesterheld & Sala (1990)* also found that grazing increases the number of individuals of dominant species. This result occurs because, in addition to the seeds themselves, livestock have a high ability to spread seeds (*Rosenthal, Schrautzer & Eichberg, 2012*), create more favorable habitat opportunities for seed germination, and reduce individual damage caused by competition within or between species. This phenomenon suggests that compared to the *S. breviflora* population under the non-grazing treatment, the heavy grazing treatment adopts a mechanism of accelerating the population renewal rate to maintain the relative stability of the population size. Further analysis found that in 2015 and 2017, there was no significant difference in the number and proportion of *S. breviflora* individuals between the heavily grazed and non-grazed treatments. In arid and rainless desert steppes, extreme precipitation events during the growing season may be triggers that impact the number of dominant populations. Consistent with this pattern, in 2014 (precipitation in May–July = 51.7 mm) and 2016 (precipitation in May–July = 122 mm), the population of heavily grazed *S. breviflora* represented a smaller proportion of the plant community than that of non-grazed *S. breviflora*, and in 2016, the density of non-grazed *S. breviflora* seedlings and juveniles was significantly higher than that of heavily grazed *S. breviflora* seedlings and juveniles.

This study explored the potential maintenance mechanism of dominant populations from a new perspective of seed germination rate. However, unfortunately, the chemical composition changes of awns under different treatments were not analyzed, which means that the biochemical perspective was not included. Additionally, the density of *S. breviflora* individuals at different growth stages was not investigated from the beginning of this study, but from 2015; thus, the typical drought year of 2014 was missed.

## CONCLUSION

The results suggested that heavy grazing reduced the phenotypic characteristics of *S. breviflora* seeds but did not significantly reduce the seed germination rate, and the awns

did not inhibit seed germination. Compared with the no grazing and heavy grazing treatments, *S. breviflora* seeds made full use of the spreading functions of awns and adopted the ecological strategies of maintaining the seed germination percentage and "opportunism" to ensure population maintenance.

## ACKNOWLEDGEMENTS

We thank Guopeng Liu for his detailed graph (Fig. 1). We also gratefully acknowledge the students from Inner Mongolia Agriculture University for their help with this work.

### Funding

This research was supported by the Key Laboratory of Grassland Resources, Ministry of Education P.R. of China (Grant No. IRT-17R59), the National Natural Science Foundation of China (31460126). The funders had no role in study design, data collection and analysis, decision to publish, or preparation of the manuscript.

### Grant Disclosures

The following grant information was disclosed by the authors:
Key Laboratory of Grassland Resources, Ministry of Education P.R. of China: IRT-17R59.
National Natural Science Foundation of China: 31460126.

### Competing Interests

The authors declare that they have no competing interests.

### Author Contributions

- Wenting Liu conceived and designed the experiments, performed the experiments, analyzed the data, contributed reagents/materials/analysis tools, prepared figures and/or tables, authored or reviewed drafts of the paper, approved the final draft.
- Zhijun Wei conceived and designed the experiments, performed the experiments, analyzed the data, contributed reagents/materials/analysis tools, authored or reviewed drafts of the paper, approved the final draft.
- Xiaoxia Yang conceived and designed the experiments, performed the experiments, authored or reviewed drafts of the paper, approved the final draft.

### Data Availability

  The raw measurements are available in File S1.

### Supplemental Information

Supplemental information for this article can be found online at http://dx.doi.org/10.7717/peerj.6654#supplemental-information.

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
