# Peer review of "Maintenance of dominant populations in heavily grazed grassland: Inference from a Stipa breviflora seed germination experiment"

_PeerJ, doi:10.7717/peerj.6654_

## Round 0.1 · original submission · Major Revisions

Rev#2 made several major remarks that need to be carefully addressed.
Some of them concern the request of more details on seed germination and possible related consequences.

I particularly agree with the first observation of the reviewer when they say "1) Seeds of Stipa breviflora generally dispersed in June, and germinated in following season, thus it is important to test new harvest seed germination traits rather than 1 year stored seeds, in particularly when you want to infer their ecological significance".

I think that this point could be hardly solved without additional research. So please, pay particular attention when rebutting this remark and/or consider to reorganize the results-discussion for minimizing its related issues.

·

Basic reporting

This article examine survival strategies associated to seed morphological and physiological characteristics in Stipa brevifolia.

The objectives and hypothesis are important for the area of research
Please, put in chronologic order cites in the manuscript (for example, line 33).

Experimental design

Materials and Methods
Fails to detail the conditions of temperature and light in the tests of germination (lines 105-110).
Please, explain the concept of “germination rate”, it is not clear if you consider germination percentage or velocity of germination (line 116).
I would like to read the previous research that permit me understand the addition of awns to the culture medium for the germinations test (lines 116-126)
In addition, you must justified the election of cabbage seed to verify the factors affecting germination.

Validity of the findings

Results
The presentation is clear. Although are confuse the use of abbreviation for name the treatments.

Discussion
Is correct and the Conclusions are well defined.
Please, to correct references (line 197; 205 De Miguel et al; etc) and number of figures (line 225, is Fig 2 or 3).

References
The references list are incomplete. For example “Liu et al, 2018a” (line 60) did not appears.
Furthermore, some cites are erroneous write, because of first appearance the name instead the surname. For example “Martin & Sala 1990 (line 260) instead: Oesterheld & Sala, 1990.

Reviewer 2 ·

Basic reporting

No comment.

Experimental design

There are some flaws in experiment design: 1) Seeds of Stipa breviflora generally dispersed in June, and germinated in following season, thus it is important to test new harvest seed germination traits rather than 1 year stored seeds, in particularly when you want to infer their ecological significance; 2) More detailed information about seed germination needed to be added, e.g. germination temperature, light condition; 3) grass species are generally formed as bunch which may contain seedlings, junveils, and adults together, I am not sure if it is reasonable to count seed age based on the basel diameter.

Validity of the findings

This is an interesting subject. However, I am wonder if the current experiment design and results can address the question proposed in this study. 1) firstly, all the conclusion in this study are based on seed germination traits of stored seeds, however, this is not consistent with true condition in the field. 2) awn have been showed have a large effect on seed germination, what is the potential mechanism? It is so hard to convice me that grazing changed the awn chmeical composition and thus affect seed germination. Further, awn of Stipa species is known to affect seed dispersal and its ancor to soil, how this structure affect seed germination in the field? 3) the author claimed that high germination rate is a potential adaptive mechanism to heavy grazing, how? If seeds try to avoid grazing stress, it seems to they need a longer time to escape the current habitat, and delayed germination and seed dormancy seems to be more adaptive.

Additional comments

No more comments.

---

## Round 0.2 · Minor Revisions

I have now reconsidered your revised version of the manuscript, together with rebuttal letter to reviewers' remarks. I appreciate the considerable effort you did for clarifying some points raised by reviewers. I have to say that the framework of your experiment is now more clear to me in light of these replies. I understand that most of information is already in the text, but I suggest to expand it further in order to support the reader who can encounter the same misunderstanding posed by the reviewer.

In practice, my suggestion is to add part of your responses to reviewer in the appropriate part of the text (and in an appropriate style).
In particular, I am referring to

- 1. Response to comment: (There are some flaws in experiment design: , etc. and to
- 5. Response to comment: (2) awn have been showed have a large effect on seed germination, etc.

While it may result a bit colloquial, I think that this solution may highlight the serious effort you did for planning the experiment and may also support researchers who would like to replicate your design.

---

## Round 0.3 · accepted · Accept

I think that your changes regarding the formulation of the hypothesis and the explanation of the underlying mechanisms have met reviewer's requirements. Your manuscript is now suitable for publication on PeerJ.

#